# Macrophage Modification Strategies for Efficient Cell Therapy

**DOI:** 10.3390/cells9061535

**Published:** 2020-06-24

**Authors:** Anastasiya S. Poltavets, Polina A. Vishnyakova, Andrey V. Elchaninov, Gennady T. Sukhikh, Timur Kh. Fatkhudinov

**Affiliations:** 1National Medical Research Center for Obstetrics, Gynecology and Perinatology Named after Academician V.I. Kulakov of Ministry of Healthcare of Russian Federation, 4 Oparina Street, Moscow 117997, Russia; a_poltavets@oparina4.ru (A.S.P.); elchandrey@yandex.ru (A.V.E.); g_sukhikh@oparina4.ru (G.T.S.); 2Department of Histology, Cytology and Embryology, Peoples’ Friendship University of Russia, 6 Miklukho-Maklaya Street, Moscow 117198, Russia; fatkhudinov@gmail.com; 3Department of Histology, Pirogov Russian National Research Medical University, Ministry of Healthcare of The Russian Federation, 1 Ostrovitianov Street, Moscow 117997, Russia; 4Scientific Research Institute of Human Morphology, 3 Tsurupa Street, Moscow 117418, Russia

**Keywords:** macrophages, cell therapy, genetic modification, inflammation, polarization

## Abstract

Macrophages, important cells of innate immunity, are known for their phagocytic activity, capability for antigen presentation, and flexible phenotypes. Macrophages are found in all tissues and therefore represent an attractive therapeutic target for the treatment of diseases of various etiology. Genetic programming of macrophages is an important issue of modern molecular and cellular medicine. The controllable activation of macrophages towards desirable phenotypes in vivo and in vitro will provide effective treatments for a number of inflammatory and proliferative diseases. This review is focused on the methods for specific alteration of gene expression in macrophages, including the controllable promotion of the desired M1 (pro-inflammatory) or M2 (anti-inflammatory) phenotypes in certain pathologies or model systems. Here we review the strategies of target selection, the methods of vector delivery, and the gene editing approaches used for modification of macrophages.

## 1. Development of Macrophages

Macrophages constitute a heterogeneous cell population representing innate immunity. Discovered at the end of 19th century by Ilya Mechnikov [1], macrophages have been identified in all tissues. Their chief competences are phagocytic activity and antigen presentation. Macrophages continuously monitor their microenvironments for the presence of pathogens, unfit cells, debris, and toxic metabolites, and release a variety of active substances including growth factors and cytokines [2]. Human macrophages express a number of markers including CD14, CD16, CD68, CD163, CD11b, CD86, and CD206 [3,4]. By contrast with many other cell types, macrophages cannot be traced to a single origin. The modern concept of macrophage origin includes three developmental sources of these cells, which correspond to the three generations of hematopoietic stem cells [5,6]. The first generation develops from the extraembryonic yolk sac posterior plate mesoderm in the blood islands [7,8]. These cells apparently give rise to the microglia of the central nervous system [9]. The second wave of hematopoietic progenitors, which develops from the yolk sac hematogenic endothelium, is called erythro-myeloid precursors. After the onset of blood circulation, these cells colonize the embryonic liver. Erythro-myeloid precursors give rise to granulocyte, monocyte and macrophage lineages [10,11]. The third generation of hematopoietic cells is derived from endothelium in the aorto-gonado-mesonephral zone; these cells colonize the fetal liver where they establish hematopoiesis and the red bone marrow where they produce the bone marrow hematopoietic stem cell lineages [6]. During embryogenesis, macrophages in organs are predominantly represented by cells of the second and third generations, while postnatal development is marked by increasing percentage of macrophages derived from the third generation [12]. The exact origin of resident macrophages in certain organs requires detailed investigation. 

## 2. The M1/M2 Paradigm

In the 20th century, macrophages (and the innate immunity in general) were largely neglected due to the dominance of adaptive immunity as the main research focus. Recent decades have brought a renaissance to the innate immunity research. The attitude to macrophages, previously considered simply as scavengers of cellular debris and other wastes of the normal functioning or infections, has also changed. The shift occurred by understanding the remarkable plasticity of macrophage phenotypes. Currently, macrophages are thought to play a major role in the regulation of tissue and organ homeostasis, as well as in autoimmune diseases, atherosclerosis, and cancer [13]. 

The most-discussed current classification of macrophages is based on the M1/M2 paradigm, which is related to their pro- and anti-inflammatory properties [14]. Proposed in the early 21st century, the M1/M2 paradigm states that macrophages can switch their phenotypes from the pro-inflammatory M1 to the anti-inflammatory M2 and vice versa, depending on the needs of the microenvironment, or maintain the naïve state M0 in the absence of external signals [15]. Exogenous inducers of macrophage polarization have been extensively studied. The classical activation of macrophages is promoted by lipopolysaccharide (LPS), interferon gamma (IFN-γ), and granulocyte-macrophage colony-stimulating factor (GM-CSF). The resulting classical (M1) phenotype is characterized by expression of TLR-2, TLR-4, CD80, and CD86. M1 macrophages secrete pro-inflammatory cytokines (interleukins IL-1β, IL-12, IL-18 and IL-23, and tumor necrosis factor alpha TNF-α) that modulate the Th1-mediated antigen-specific inflammatory reactions (Table 1) [16,17]. M1 macrophages have also been demonstrated to enhance the expression of inducible nitric oxide synthase (NOS2 or iNOS) to facilitate the production of NO from L-arginine [18]. 

Activation of macrophages towards M2 phenotypes can be induced by antigen-antibody complexes, invading helminths, complement system components, apoptotic cells, interleukins (IL-4, IL-13, and IL-10), and transforming growth factor beta (TGF-β). Activation with these inducers drives macrophages towards the increased secretion of IL-10 and reduced secretion of IL-12 typical of the M2 phenotypes [19]. M2 macrophages show diverse gene expression signatures, and distinct M2a, M2b, M2c, and M2d macrophage subpopulations have been identified by transcriptome analysis. The corresponding M2 phenotypes are generally characterized by high levels of mannose receptor CD206 and scavenger receptor CD163. Arginase 1, which converts arginine into ornithine—an important building block for collagen synthesis, is a relevant marker of M2 macrophage polarization in rats and mice. 

The M1/M2 paradigm is often criticized as too simplistic. A recent study by Specht et al. demonstrated that mature macrophage populations exhibit a continuum of phenotypes even in the absence of inducers [20]. Single-cell proteomic analysis of randomly picked macrophage samples revealed the coexistence of M1 and M2 signatures within the same culture in the absence of polarizing stimuli. However, the M1/M2 paradigm conveniently reflects the most phenotypically distant (polar) differentiation states of macrophages, and the terminology has caught on and is often implied in research. However, for in vivo studies, the M1/M2 dichotomy should be used with caution and possibly replaced with the terms pro-inflammatory/pro-regenerative, respectively. 

## 3. Macrophages as Perfect Cells for Reprogramming 

Regulation of cell homeostasis under pathological conditions by modulating macrophage behaviors in the affected tissues and organs is an important prospect of modern biomedicine. Such interventions may involve in situ or ex vivo reprogramming (the latter implies autologous transplantation of the modified macrophages). In proliferative diseases, the evoked M1 polarization of macrophages will stimulate the processes of inflammation and cell death, whereas in the case of inflammatory diseases the evoked increase in the M2 macrophage content will stimulate the processes of regeneration, angiogenesis, and extracellular matrix remodeling. The relevance of macrophages for cell therapy as either a target or an agent is undeniable. The use of macrophages as a therapeutic tool is restricted by the lack of safe and efficient approaches for their reprogramming. Overcoming this problem requires further investigation of the roles of particular genes in macrophage functionalities. The ultimate goal is the development of efficient approaches for genetic modification of these cells in order to suppress or boost the expression of certain genes for obtaining stably polarized M1 or M2 species. The resulting cell lineages with specifically altered phenotypes can be used as a platform for studying gene interactions and signaling pathways to afford reliable targets for drug development. 

This review is focused on the use of macrophages in experimental and clinical research and the methods for specific alteration of gene expression in macrophages, including the controllable promotion of the desired M1 or M2 phenotypes in certain pathologies or model systems. The majority of studies in this field employ immortalized human or murine cell lines which can be modified more straightforwardly and reproducibly than primary cultures. At the same time, the continuous upgrade of methods for the delivery of genetic constructs to cells will certainly allow modification of the primary macrophage cultures for autologous transplantations. 

### 3.1. The M1 Macrophage Applications 

The classically activated (M1) macrophages show pro-inflammatory activity. They express MHC class II molecules on their surface, which defines their participation in the immune response and facilitates their interactions with other immune cells. M1 macrophages are also implicated in the recognition and elimination of cancer cells; they have been shown to suppress tumor growth and metastasis. In cancer patients, survival is frequently associated with elevated numbers of M1 macrophages. In this aspect, M1 macrophages are of great interest, and many studies are aimed specifically at obtaining this phenotype for therapeutic purposes. Understanding the molecular principles of M1 polarization will allow using it in anticancer therapies. This view is supported by a number of studies. For instance, Zhang et al. used modified messengers of two pro-inflammatory polarization markers, IRF5 and IKKβ, for the in vivo macrophage reprogramming in mouse model. The mRNA molecules packaged into polymer nanoparticles were administered to mice by intravenous or intraperitoneal injections. The in vivo transfected cells showed high anti-tumor activity, with significant regression of ovarian tumors in 40% of the cases. Besides, the IRF5 and IKKβ overexpression-mediated macrophage reprogramming into pro-inflammatory phenotypes alleviated metastasis and exerted positive therapeutic effects for lung tumors and gliomas [23]. 

The mainstream molecular cascades of macrophage polarization are supported by a number of significant ancillary pathways. For instance, chloroquine, a common antimalarial, was shown to exert anti-tumor effects. Chloroquine promoted the M2-to-M1 macrophage reprogramming by increasing pH within the lysosomal compartment of macrophages with a concomitant increase in the efflux of Ca^2+^ via the mucolipin channels (*MCOLN1*). This type of channel is engaged in the activation of p38 and NF-κB, which are key sentinels in M1 polarization. Simultaneous activation of the transcription factor EB (*TFEB*) facilitates attraction of other immune cells for stabilization of the acquired phenotype [24]. Proper assessment of various factors that contribute to the polarization process will help efficiently and reproducibly direct it towards M1 macrophage phenotypes for anti-tumor therapies. Certain anti-tumor drugs targeted at macrophages show high efficiency (for instance, Paclitaxel interferes with the ovarian tumor growth by shifting macrophage polarization from M2 to M1) [25]. 

Thus, pro-inflammatory (M1) macrophages are vitally important participants of tumor suppression. Obtaining them as pure cultures or in vivo clusters is highly relevant for the development of anticancer drugs and therapies. Recent studies confirm that macrophages really matter, e.g., high levels of M1 macrophage infiltration and low levels of M2 macrophage infiltration of tumor islets in non-small cell lung cancer patients have been associated with increased survival [26]. It should be also emphasized that the M1 polarization studies generate new fundamental knowledge about the mechanisms of inflammation. 

### 3.2. The M2 Macrophage Applications 

The alternatively activated (M2) macrophages have pronounced anti-inflammatory properties. These cells are active phagocytes implicated in the extracellular matrix remodeling and angiogenesis. M2 macrophages suppress immune response and are considered tumorigenic. It should be noted that tumor-associated macrophages generally exhibit M2 phenotypes. Basic mechanisms of M2 polarization, as well as its principal markers, are being investigated in order to counteract the devastating participation of these cells in the tumor growth and metastasis. A recent article by Lee et al. features a prospective anticancer therapy with melittin (a peptide that binds to CD206 at the surface of tumor-associated macrophages) fused to a pro-apoptotic peptide that triggers mitochondrial death upon internalization [27]. 

The opposite strategy of forcing macrophages into M2 phenotypes may be applied for the treatment of atherosclerosis [28]. LAIR1 (CD305) is an inhibitory collagen receptor expressed by certain blood cells, e.g., by monocytes. As demonstrated by Yi et al., silencing of *LAIR1* in immortalized human monocytes facilitates M2 polarization indicated by increased expression of ARG1 and CD206 [29]. This study identifies *LAIR1* as a promising therapeutic target in atherosclerosis and related disorders. Another study demonstrated the involvement of histone deacetylases (HDAC) in the early recruitment of reparative CD45+/CD11b+/CD206+ macrophages to the heart after myocardial infarction and its positive correlation with the ventricular function and remodeling [30]. A study by Cao et al. demonstrates that in the histone deacetylase 9 knockout mice (*Hdac9*^-/-^), macrophages are inclined towards M2 phenotypes and show decreased expression of the pro-inflammatory M1 markers [31]. The interest of the scientific community in M2 polarization was initially due to its association with tumorigenesis. Macrophages of this phenotype represent a potential target for genome editing and reprogramming in various diseases, in connection with their anti-inflammatory functions. 

The role of M2 macrophages in tissue repair is prominent. They have been shown to promote angiogenesis and tissue remodeling by releasing a palette of factors (VEGF, TGF-β, FGF, etc.) [32]. Incidentally, increased numbers of anti-inflammatory macrophages correlate with excessive collagen deposition in patients with kidney fibrosis, lung fibrosis, or sclerotic skin lesions [33]. These findings highlight the important role of M2 macrophages in tissue repair. 

## 4. The Role of the Monocyte-Macrophage System in Disease, or Why Should We Reprogram Macrophages? 

### 4.1. Inflammatory Diseases

Macrophages play a dual role by promoting inflammatory reactions on one hand and supporting tissue regeneration on the other hand. In patients with genetic syndromes (e.g., muscular dystrophies) or systemic disorders (obesity, autoimmune reactions, tumors), macrophages become involved in exacerbation of fibrosis, atherosclerosis, tumor growth, etc. (Table 2). In the acute phase of inflammation, macrophages become classically (M1) activated. Along with neutrophils, they scavenge dead cells and tissue debris at the site of injury and release various pro-inflammatory cytokines that stimulate alteration and exudation. By the end of the acute phase, the number of alternatively activated (M2) macrophages at the focus of inflammation increases, while the number of M1 cells is decreasing, which indicates the involvement of the M2 cells in the transition to the productive stage of inflammation and regeneration [34,35]. The outcome of the inflammatory reaction (regeneration or fibrosis) correlates with the timely switching of macrophage polarization from M1 to M2 [36]. The M1 macrophages are extremely important for the initial stages of inflammation: they ensure debridement of the site by phagocytosis and promote chemotaxis of other immune cells by expressing pro-inflammatory cytokines, notably IL-1β, IL-6, IL-8, TNF-α, and IFN-γ. The switch towards M2 phenotype occurs on days 3–7: the macrophages acquire anti-inflammatory properties and start secreting IL-10 and TGF-β to promote regeneration and angiogenesis [37].

Misbalanced M1 and M2 macrophage polarization in the human body may cause chronic illnesses. With the rise of M1/M2 paradigm, the course of typical pathological processes has been reassessed in terms of macrophage polarization. For instance, pro-inflammatory macrophages are abnormally overrepresented in multiple sclerosis and encephalomyelitis, and the shift to M2 phenotype significantly reduces clinical symptoms suggesting that increasing the ratio of anti-inflammatory cells could be a successful therapeutic strategy [38]. A similar shift towards M1 polarization in rheumatoid arthritis is implicated in the excessive osteoclastogenesis [39], and its suppression may help alleviate the progressive deterioration of bone tissue in this disease. 

Contribution of macrophages to the central nervous system pathologies is evident as well. Although myelin phagocytosis has been shown to facilitate the M2 polarization in vitro, macrophages in a damaged spinal cord are strongly inclined towards M1 polarization, which interferes with the neural tissue recovery [40]. In Alzheimer’s, microglial macrophages of the brain react to the amyloid β (Aβ) peptide by releasing pro-inflammatory factors that promote their own phagocytic activity. The clearance of Aβ peptide by phagocytosis may slow down the disease; on the other hand, the pro-inflammatory milieu can exacerbate the condition by a positive feedback to the activity of M1 macrophages [41]. In Parkinson’s disease, dysregulation of the JAK/STAT pathway in myeloid cells has been shown to promote neuroinflammation that forces macrophages into M1 phenotypes and exacerbates the neuronal damage [42]. 

The M2 macrophage polarization may also contribute to pathological processes, as proved to be the case in certain parasitic diseases. Immunopathogenesis of the chronic skin leishmaniasis is associated with the increased polarization of macrophages towards M2 phenotypes. It is important to note that existing drugs against leishmaniasis promote the reprogramming of M2 macrophages into M1 phenotypes [43]. Vidal et al. demonstrated upregulation of TGF-β synthesis with further collagen accumulation facilitated by an increase in the activity of the alternatively activated macrophages in a mouse model of Duchenne muscular dystrophy [44]. The asthma-associated fibrosis is also related to a shift toward M2 polarization [45]. Hepatic cirrhosis is associated with a significant enhancement of the M2 signatures in the liver [46]. Similarly, the yeast-like fungi *Pneumocystis murina* infections in mice involve polarization of alveolar macrophages into M2 phenotypes [47]. 

These findings indicate that controllable alterations of macrophage phenotypes can provide therapeutic effects for a number of inflammatory and autoimmune disorders.

### 4.2. Proliferative Diseases

Tumor-associated macrophages (TAMs) are highly relevant in modern biomedicine. TAMs constitute a distinct subpopulation of immune cells in tumor microenvironments. These cells may originate from embryonic sources (similarly with resident macrophages) or differentiate from circulating monocytes [48]. TAMs play important roles in tumor growth and metastasis; they are implicated in chemoresistance. The M1/M2 stratification of TAMs is controversial and should be applied with caution, although the M2-like TAM phenotypes have been generally associated with dismal prognosis [49]. Macrophages with the anti-inflammatory M2 phenotypes are considered tumorigenic as they facilitate angiogenesis, extracellular matrix remodeling, tumor progression and metastasis, and therefore represent a potential target for anticancer therapies.

High densities of TAMs, estimated by expression of CD68 and CD163, correlate with poor clinical outcomes for breast cancers, thyroid cancers, head and neck cancers, hepatic, urinary, renal, pancreatic, ovarian, endometrial, oral and pulmonary neoplasms, vascular tumors, and Hodgkin lymphomas [50,51,52]. A number of studies reveal tumorigenic properties of TAMs: they promote angiogenesis, inhibit the anti-tumor immunity (e.g., the T cell-mediated cytotoxicity) and secrete the extracellular matrix remodeling factors that increase the tumor cell motility and invasiveness. Significant transcriptomic distinctiveness of brain TAMs as compared to normal microglia have been demonstrated; this finding illustrates the remarkable in situ plasticity of macrophages [53]. Targeted elimination of TAMs is a promising concept for anticancer drug development. Importantly, the origin and heterogeneity of TAMs, and their specific contribution to pathogenesis, may depend on the tumor identity. For instance, selective depletion of resident TAMs in the ductal pancreatic adenocarcinoma therapies appears highly feasible, as progression of this particular tumor is dependent on these cells [48]. Although targeted elimination of TAMs can be advantageous, their total exclusion would critically undermine the effectiveness of macrophage-dependent therapeutic approaches, including the PD-1 and CTLA-4 targeted immunotherapies, which may also function through a direct effect on macrophages [54,55], and the use of the monoclonal antibody antineoplastics. As the elimination of TAMs is ambivalent, it may be a finer idea to promote anti-tumor response from the immune system by cajoling TAMs into M1 phenotypes. The possibility of boosting anti-tumor activity of macrophages by evoking M1 polarization of TAMs is being intensively explored [56,57].

## 5. The Existing Approaches for Macrophage Reprogramming

### 5.1. Early Attempts at Obtaining Specific Phenotypes

Isaiah Fidler is justly considered the founder of the ex vivo activated macrophage transplantations because he discovered the anti-tumor effect of the ex vivo stimulated macrophages in a mouse model of lung cancer [58]. However, a considerable number of the ex vivo macrophage reprogramming experiments carried out since the 1990s [59] revealed no significant anti-tumor effects of the reprogrammed macrophage transplantations in various cancer models [60,61]. Analysis of the literature displays methodological and technical obstacles encountered during the research. First of all, the heterogeneity of human peripheral blood monocytes was largely neglected [62,63], as it was only recently introduced as a paradigm. Secondly, the reprogramming itself was transient, and macrophages regained their original phenotypes under the influence of tumor microenvironments. Overcoming these obstacles by means of advanced molecular biology will give a new impetus to the macrophage reprogramming and help translate it into clinical practice.

### 5.2. Activation with Signaling Molecules

The existing methods for the transient reprogramming of macrophages with signaling molecules (cytokines, receptor agonists, inhibitory antibodies, etc.) are currently being translated to the clinic. Various blockers of cytokines or their cell surface receptors can be applied in order to prevent the M2-like polarization of macrophages under the influence of tumor signaling. A number of promising candidate molecules (including CSF-1 receptor kinase inhibitors, CCL2/CCR2 receptor antibodies, and VEGF inhibitors [64,65,66]) interfere with the binding of cytokines to their receptors at the macrophage surface and prevent further recruitment of macrophages to the tumor. A series of clinical trials for these drug candidates have been launched. Their downside is the severe side effects of high cytotoxicity, especially in combination with the non-selective action of certain inhibitors (e.g., antibodies to CCL2) on different cells. For instance, CCL2 inhibitors effectively suppress metastasis in preclinical models [67]; however, a break from such therapy may accelerate the breast cancer metastasis by promoting angiogenesis [68].

In their recent study, Shields et al. successfully employed a particle-based technology of macrophage transfer to induce the pro-inflammatory immune response [69]. Macrophages carrying the “IFN-γ backpacks” showed their efficacy against murine mammary carcinoma by conferring stable M1 polarization of TAMs. Important findings in this area also include the suppression of tumorigenic activity of TAMs by IFN-γ in ovarian cancer patients [70] and the significant tumor regression accompanied by increased expression of M1 markers (MHC class II and CD86) induced with CD40 agonist antibodies in a model of the ductal pancreatic adenocarcinoma [71].

After the link between the clinical effects of anti-tumor drugs and the activation of macrophages became apparent, selective targeting of TAMs turned into a mainstream. It is currently aimed at development of in vivo interventions specifically targeted at macrophages without affecting other cell populations, as the spectrum of target cells for cytokines and other regulatory molecules is wide (while the effects themselves are transient).

### 5.3. Identification of Relevant Genomic Targets

Macrophages can be genetically reprogrammed to sustain a desired phenotype. Proper selection of the target, i.e., a gene or regulatory sequence to be destroyed or edited, is essential. Due to the extreme complexity of metabolic and signaling processes, this task is very difficult. Successful strategies for selecting a genomic target can be based on either the analysis of transcriptome (or proteome) or the understanding of gene function.

Transcriptome analysis is a promising way to identify potential targets for the genetic reprogramming. Analysis of differential gene expression in a pair of samples (e.g., M1 and M2 macrophages) yields a list of the candidate genes with mRNA levels differing greatly between the samples. This approach allows identification of genomic sequences that are crucially important for the macrophage polarization. Transcriptomic data typically comprises huge arrays of differentially expressed genes, and it is vital to select the real influencers among them in order to benefit. For instance, Gerrick et al. suggested a selection of 14 key phenotype-determining genes on the basis of comparison of transcriptomes for M1 and M2 polarized macrophages [72]. Proteomic analysis as a complementary approach is highly feasible; however, Specht et al. have recently demonstrated that monocytic macrophages exhibit a mixture of M1 and M2 proteomic signatures even in the absence of polarizing stimuli [20]. Selection of targets by comparative proteomic analysis is eligible only for cells at the final stage of differentiation (confirmed by the expression of polarization markers) and requires comprehensive understanding of their biology.

Selecting target genes for modification on the basis of high-throughput molecular profiling is not a self-sufficient strategy though, as the bulk of omics data requires the structuring by biological relevance. The macrophage polarization mechanism consists of four distinct levels, or stages, that proceed under control of different signaling molecules—polarization inducers, receptors, transcription factors, and effector molecules (Table 1). The corresponding genes, or their direct regulatory partners, should be favored as targets for modification, since alterations in their expression (or expression of their synergists/antagonists) will most likely influence the balance of macrophage phenotypes in the studied system.

### 5.4. Transient Modifications

Small interfering RNAs (siRNAs) promote the elimination of transcripts in a sequence-specific manner. Despite its transience, specific inhibition of genes with corresponding siRNAs is remarkably efficient and represents a perfect system for the pilot evaluation of genomic targets.

With the use of siRNAs, it has been demonstrated that knockdown of *CYBB* in M1 macrophages in vitro leads to a significant decrease in the expression of M1 marker CD80 and decreased secretion of M1 effector molecules TNF-α and CXCL9 [72]. Gene *CYBB* encodes a subunit of cytochrome b-245 responsible for the generation of reactive oxygen species. Previous studies have shown the importance of CYBB and the NADPH oxidase complex in the TLR- and IFNγ- dependent oxidative burst and the concomitant pro-inflammatory cytokine synthesis [73,74]. The finding indicates that *CYBB* is involved in the M1 polarization signaling and its inhibition in macrophages facilitates a shift towards M2 phenotype.

A recent article by Yu et al. features STAT6 acetylation as a negative regulatory switch that restricts M2 polarization in macrophages. Knockdown of *TRIM24*, the E3-ubiquitin ligase-encoding gene that works as a positive regulator of STAT6 acetylation, induced M2 polarization in human macrophages in vitro [75].

Another study was focused on the C-X-C chemokine receptor type 4 gene *CXCR4* implicated in acute lung injury (ALI) [76], a life-threatening condition associated with increased permeability of the alveolar-capillary barrier. It is known that binding of CXCL12, which is a ligand of CXCR4 in macrophages, induced the activation of pro-survival ERK and AKT signals, while pro-inflammatory p38 and JNK signals were activated by the binding of CXCL12 to CXCR7 in macrophages [77]. In ALI, macrophages readily interact with LPS of bacterial species breaking through the barrier, which drives them towards M1 phenotype; the effect is reinforced by profuse secretion of pro-inflammatory mediators and cytokines including TNF-α, IL1β, IL6, iNOS, and the macrophage migration inhibitory factor (MIF). An RNA interference knockdown of *CXCR4* induced a decrease in IL-6 and TNF-α expression accompanied by increase in the expression of IL-10 mediated by suppression of MAPK and NF-κB signaling pathways in vitro. These data are of potential clinical value for the treatment of ALI and possibly other inflammatory conditions as well [76].

The RNA interference knockdown technique is widely used in laboratory practice. Applicable to a variety of models, this approach is much valued for being transient and safe. Importantly, it does not affect the genome integrity.

Another method for transient cell reprogramming is the transfection of cells with coding RNA molecules produced by in vitro transcription (IVT-RNA). Zhang et al. used this approach to promote overexpression of the macrophage-specific factors of anti-tumor response—interferon regulatory factor 5 (IRF5) and the IKKβ kinase (which activates IRF5 by phosphorylation) [23]. Bipolimeric nanoparticles assembled for the in vivo delivery were charged with the IVT-RNA containing coding sequences for IRF5 and IKKβ (a 5’ end modification and modified nucleotides in a number of positions were introduced to facilitate stable translation of the in vitro transcripts inside the cell while avoiding their digestion by cellular nucleases). The IVT-RNA transfection is the most straightforward means for the transient overexpression of specific sequences; with efficient and safe delivery, it can provide pronounced therapeutic effects.

### 5.5. Genome Editing

Genome editing is a limelight issue. It is also a powerful approach for studying gene functions and their relations with cell functionalities. It allows producing new cell lines with advanced properties and, ultimately, of therapeutic value. Genome editing is highly applicable to macrophages and helps to study their functions, metabolic features, migratory capacities, and intercellular interactions. Genome editing promotes stable modification and therefore enables evaluation of the long-term effects of complete or partial suppression of particular genes. A number of gene functions in macrophages of different localization have already been assessed by knockouts in murine models. For instance, transcription factor-encoding genes *Gata6*, *Runx3*, *Nr1h3*, *SpiC*, and *Pparg* were identified as principal markers of, respectively, peritoneal macrophages [78,79], Langerhans cells [80], marginal zone macrophages [81] and red pulp macrophages of the spleen [82], and alveolar macrophages [83]. These genes are required for the targeted programming of macrophages in accordance with particular therapeutic tasks and specific features of microenvironment. They ensure maturation and survival of macrophages at particular locations within the body.

Arguably the best way to get a specific knockout is mediated by CRISPR/Cas9 complexes composed of Cas9 nuclease, which introduces a double-stranded break in DNA, and guide RNA (gRNA) whose sequence determines genomic coordinates of the break. The advantage of this approach is the high specificity of the guidance in combination with the high modification efficiency.

Despite the widespread use of the CRISPR/Cas methodology in laboratory practice, the number of studies using this approach for directed macrophage polarization is still limited.

Excessive inflammation, typical for a number of disorders, is sometimes related to abnormal functioning of inflammasomes—cytosolic complexes formed by NLRP3 pathogen sensor proteins and ASC adapter proteins in combination with caspase-1. Inflammasome maturation activates caspase-1 and triggers the inflammatory response. A CRISPR/Cas9-mediated knockout of *NLRP3* in macrophages significantly reduced the amounts of active caspase-1 and generally interfered with inflammasome assembly in vivo [84]. This finding is relevant for the development of anti-inflammatory therapies.

### 5.6. Delivery of Genetic Constructs to Cells

Several techniques have been invented for transporting CRISPR/Cas complexes or other modifying agents into cells. The delivery can be mediated by viruses, chemicals, or physical factors.

The use of plasmid constructs that contain a coding sequence for Cas nuclease and a gRNA template affords high knockout efficiency [85]. The problem of transfecting macrophages with such plasmids is that macrophages recognize foreign nucleic acids and activate the immune response processes [86]. Macrophages express abundant intracellular DNases, as well as a variety of receptors and protein complexes at the cell surface, which enables them to scavenge nucleic acids released from apoptotic cells [87]. Other delivery options include the use of pre-assembled ribonucleoprotein complexes (CRISPR/Cas-RNP) instead of the plasmid, which accelerates the CRISPR/Cas operation and contributes to higher cell survival. Despite these advantages, efficient delivery of Cas protein and gRNA to macrophages is challenging.

A protocol for the in vivo delivery of CRISPR/Cas-RNP recently suggested by Lee et al. showed remarkable efficiency for the gene editing in hepatic and splenic macrophages [88]. Terminal tagging of Cas9 protein with oligo (glutamic acid) promotes its interaction with the arginine-functionalized gold nanoparticles to yield hierarchical nanocomposites that can be loaded with gRNA and used for the membrane fusion-mediated transfection. Systemic intravenous administration of such single carrier composites to mice ensured effective editing of *Pten* in >8% and >4% of hepatic and splenic macrophages, respectively. The formulation can be used as a basis for the development of selective immunotherapies for the macrophage-mediated disorders.

In complex tissue microenvironments, increased efficiency of the in vivo macrophage transfections may be accompanied by undesired delivery of the same material to other cell types. To enhance the selectivity of delivery, macrophage-specific cell surface receptors can be employed. For instance, the delivery can be mediated by polymeric complexes with mannose residues to promote the binding by CD206 mannose receptor, a C-type lectin expressed on macrophages. This approach affords very high in vivo transfection efficiencies (over 30%) [23].

It is generally recognized that the most convenient systems for the delivery of genetic constructs to cells are derived from viruses, notably lentiviruses and adeno-associated viruses (AAV). The advantages of such systems include high insert capacity (up to 6 and 4.7 kb of cargo sequence for lentiviruses and AAV, respectively), physiological mildness, low immunogenicity, and the high efficiency of delivery even to non-dividing cells.

Attempts to use lentiviral transduction for primary cultures of murine monocytes and macrophages began quite a while ago [89]. Lentiviral systems are intended to promote stable modification of transduced cells by integration of the delivered genetic material into the host cell genome. Boehler et al. used lentiviral transduction to enhance the expression of IL-10 in a primary culture of murine macrophages of bone marrow origin, as well as in the RAW264.7 cell line. The enhanced production of IL-10 ensured stable M2 polarization even in those macrophages that initially exhibited M1 phenotypes [90].

A study by Yang et al. was focused on the role of stress-inducible protein Sestrin2 (*Sesn2*) in the control of inflammation in myocardial infarction (MI) [91]. The authors demonstrated a suppressive role of the lentivirus-mediated overexpression of Sestrin2 in pro-inflammatory macrophages in the control of cardiac inflammation after MI both in vitro and in vivo. Adoptive transfer of the Sestrin2-overexpressing macrophages immediately before MI stimulated the anti-inflammatory response and cardiac tissue repair in mouse model.

The role of microRNA miR-99a in the activation of bone marrow-derived murine macrophages was assessed by Jaiswal et al. [92]. Following lentiviral transduction with a miR-99a template, the cells were stimulated with the LPS/IFN-γ cocktail for M1 polarization. Overexpression of miR-99a significantly abolished the effect of stimulation, as indicated by reduced expression of M1 markers iNOS, Mcp-1 and Il1β, and enhanced expression of M2 markers Ym-1, Arg1 and Pparγ, compared with the control.

M2 polarization can be mediated by microRNAs and other agents produced by non-immune cells. For instance, lentiviral transduction of the triple-negative breast cancer cell line MDA-MB-231 with a miR-200ab/c template and subsequent co-culturing of the transduced cells with macrophages caused M2 polarization of the latter; the mechanism involved upregulation of PAI-2 (plasminogen activator inhibitor type 2) and IL-10 by miR-200 [93]. The study underscores the phenotypical plasticity of tumor-associated macrophages and the major influence of intercellular signaling on their polarization.

Adeno-associated viruses (AAV) are low-immunogenic DNA vehicles of lower insert capacity compared with lentiviral systems. The advantage of AAV systems is the ability to provide high transient expression of the delivered sequence without its integration into the host cell genome.

The role of p38MAPK/SGK1 signaling pathway in the M2 polarization of murine macrophages was studied by Li et al. [94]. Using the AAV-mediated delivery, the authors demonstrated that p38MAPK/SGK1 signaling is crucial for the IL-4 induced M2 polarization. Transduction of the cells with a template for short hairpin RNA (shRNA) reduced the levels of M2 markers Arg1, Ym1, Fizz1, and CD206 by inhibiting translation of p38 and SGK1 messengers in a sequence-specific manner.

Do et al. demonstrated the important role of miR-511-3p in the inhibition of inflammatory response and M2 polarization of murine macrophages [95]. The AAV-mediated overexpression of miR-511-3p was associated with reduced expression of M1 markers Il1β, Il6 and iNOS, and increased expression of M2 markers Arg1, Fizz1 and Chi3l3.

Zhao et al. used the AAV-mediated delivery to assess the role of IL-34 in the inhibition of liver transplant rejection in rat model [96]. Overexpression of *Il34* promoted anti-inflammatory polarization of Kupffer cells in vivo and alleviated the liver transplant rejection.

In general, it should be admitted that viral delivery of nucleotide sequences to immortalized cell lines, primary monocyte-macrophage cultures or in vivo macrophage populations is becoming increasingly common.

### 5.7. The Induced Pluripotent Stem Cell Technologies

The discovery of induced pluripotent stem cells (iPSCs) was of primary importance to regenerative and translational medicine. A research team in Japan studied 24 candidate factors presumably related to the cell pluripotency and identified a set of four, Oct3/4, Sox2, c-Myc and Klf4, as necessary and sufficient for reprogramming fibroblasts into pluripotent stem cells [97].

This discovery gave rise to a plethora of repair-facilitating approaches. Under in vitro conditions, iPSCs can be differentiated into a variety of cell types including macrophages. It is important to note that cultured macrophages, obtained from primary cultures of peripheral blood cells or differentiated from stem cells, proliferate slowly, which makes them a difficult object for genome editing due to the low modification efficiency and poor cell survival. Editing the iPSC genome and obtaining macrophages from the modified iPSCs turned out to be a smart alternative.

Gupta et al. reported an iPSC-derived macrophage-based model for studying reverse cholesterol transport—a complex process by which cholesterol is transported from peripheral tissues to the liver for conversion or excretion. The macrophages were modified by CRISRR/Cas9-mediated double-stranded break in DNA causing a frameshift in *ABCA1* (a gene encoding the ATP-binding cassette transporter that works as cholesterol efflux pump). The modified macrophages exhibited decreased cholesterol efflux, concomitant metabolic impairments, and increased production of pro-inflammatory cytokines [98].

The ubiquitin-specific peptidase-encoding gene *USP18* contributes to interferon signaling in macrophages. This gene was successfully knocked out in iPSCs that were subsequently converted into macrophages. The *USP18* expression is induced during HIV-1 infection of macrophages, and inactivation of this gene drastically inhibits replication of the virus. The iPSC-derived *USP18*-deficient macrophages showed significant reinforcement of interferon type 1 signaling associated with the enhanced anti-HIV response [99].

Obtaining macrophages from iPSCs is a rapidly developing area. Obtaining iPSCs from macrophages is possible as well. Although the first iPSCs were obtained from fibroblasts, now some protocols afford similar pluripotent cells from circulating monocytes [100]. Compared with the sampling of skin fibroblasts from a donor, the collection of peripheral blood is less invasive. Currently, iPSCs from monocytes are most often generated by using Sendai virus-based systems proposed and consistently upgraded by the team of M. Nakanishi [101,102].

As an option, macrophages or their precursors can be modified with the use of engineered transcription activator-like effector nucleases (TALEN). Hereditary pulmonary alveolar proteinosis (herPAP) is a rare disease associated with the inability of alveolar macrophages to purify the alveolar airspace from the surface-active phospholipids. This life-threatening condition is caused by a defect in *CSF2RA* gene that encodes a subunit of GM-CSF receptor known as CD116. Due to the limited success of conventional therapies for herPAP, the focus eventually shifted to gene therapy. It was suggested to use the genetically edited pulmonary macrophages derived from the herPAP-specific iPSCs (herPAP-iPSC) by TALEN technology. A line of the herPAP-iPSCs cells stably expressing *CSF2RA* was obtained by the TALEN-promoted insertion of codon-optimized *CSF2RA* cDNA in AAVS1 locus (a validated safe harbor site for transgenes). The modified cells were subsequently differentiated into human macrophages with characteristic morphology, phagocytic and secretory activities typical of macrophages, the CD45 + CD14 + CD11b + CD163 + CD19- surface phenotype, and functional expression of *CSF2RA* confirmed by functional tests (STAT5 phosphorylation and GM-CSF internalization) [103]. This study highlights the feasibility of the TALEN-mediated gene transfer for the production of functionally corrected monocytes and macrophages for gene therapy.

### 5.8. In Vivo Models for Genome Editing

Macrophage studies are clearly not limited to in vitro models. The use of transgenic animals greatly facilitates our understanding of macrophage functionalities. For instance, macrophages modified to overexpress the gene for angiotensin-converting enzyme (ACE) conveyed resistance to lymphoma and melanoma in mouse model; the protective effect was accompanied by increased production of IL-12 and nitric oxide and decreased production of IL-10 [104]. The use of LysM-LG transgenic mice, whose macrophage express luciferase, allowed tracing of the anti-tumor activities of pro-inflammatory macrophages by fluorescence. After intraperitoneal injection of an M1 polarization inducer, the animals with metastatic ovarian cancer showed increased activity of peritoneal macrophages. This model provides a non-invasive way to trace the recruitment and activation of macrophages during the initiation and progression of disease. It can be a useful tool for visualization and monitoring of various drugs targeted at macrophages [105].

## 6. Conclusions

Genetic reprogramming of macrophages is an important issue of modern molecular and cellular medicine. Controllable activation of macrophages in vivo and in vitro will provide effective treatments for a number of inflammatory and proliferative diseases (Table 3).

The ubiquitous presence of macrophages in organs and tissues makes them universal candidates for the use in cell and gene therapies. Relevance of these cells as therapeutics certainly depends on the possibility of correcting their transcriptome, optionally by genome editing, and reprogramming them in order to restore or activate certain functions involved in stimulating tissue regeneration or anti-tumor activity. Each article cited in this review highlights the specificity and uniqueness of the effects for a particular method; however, put together, these works demonstrate the striking versatility of the use of innate immunity cells in studying and treatment of various diseases.

Genetic modification of monocyte and macrophage cultures is frequently complicated by their limited proliferative capacity, as even with good transfection efficiencies, the successfully modified cells eventually die out. The advent of the iPSC technologies provides a solution to this obstacle. The induced pluripotent cells are highly proliferative, and the modified (edited) cells can be easily manipulated to produce any number of colonies of identical cells for the analysis and subsequent differentiation into specific lineages. Besides, obtaining induced pluripotent cells from blood monocytes of a patient, as the least invasive method for generation of autologous cells, can afford a variety of lineages for studying their properties and future use in cell therapies.

Differential analysis of transcriptomes and proteomes reveals significance of certain genes, e.g., *CYBB* [72], for the M1/M2 macrophage polarization. The choice of molecular target depends on the intended reversibility of the modification (temporary or stable). For safety reasons, the transient correction of gene expression in transplanted cells is preferable. On the other hand, stable genetic modifications (notably by using the CRISPR/Cas9 approach) will facilitate creation of disease-specific test systems and models for the development of advanced cell therapies.

The complexity of maintaining the terminally differentiated cell cultures characterized by specific expression of numerous cell surface receptors and proteins should be considered as well. The low transfection efficiencies observed for such cells can be improved by applying the viral transduction and cell type-specific delivery approaches. A notable example is the use of pre-assembled polymer complexes functionalized with the ligands to cell surface receptors; such targeted delivery systems are suitable for in vivo applications owing to their high selectivity towards a particular cell type. The choice of transfection method for in vitro applications is largely determined by the purpose of the obtained modified cultures.

Development of the treatment strategies featuring reprogrammed or modified macrophages became the next logical step in the evolving practice of cell immunotherapy after the approval of CAR-T cell therapies by FDA. A recent study by Klichinsky et al. features genetically engineered human macrophages that express chimeric antigen receptors to direct their phagocytic activity against tumors, highlighting cell therapy with modified macrophages as a rapidly developing field [106]. Other studies cited in this review illustrate the global progress in this area and inspire the confidence that personalized treatment of cancer and other diseases with autologous monocytic cells will become a reality in the near future.

## Figures and Tables

**Table 1 cells-09-01535-t001:** The levels of macrophage polarization control, with the lists of key signaling molecules and the feasible tactics of genome target selection (based on [18,21,22]).

Levels of Control	Key Signaling Molecules		Examples of Tactics
	M1 Polarization	M2 Polarization	
Surface and matrix markers	TLR-2, TLR-4, CD80, CD86, iNOS, MHC-II	CD206, CD163, CD209, FIZZ1, Ym1/2, Arginase	search for blockers, antibodies
Transcription factors	NF-kB, STAT1, STAT5, IRF1, IRF5	STAT6, IRF4, JMJD3, PPARδ, PPARγ	search for genetic blockers, search for the genes encoding proteins that regulate (activate or repress) the expression of transcription factors
Receptors	IFNGR1/2, CSF2Rα, IL1R, TLR4, CD163, CCR7	FcyR, IL4Rα, IL10R, IL1R, TLR4	search for blockers, antibodies, search for the receptors that trigger the reverse polarization process
Chemokines	CXCL1, CXCL3, CXCL5, CXCL8, CXCL9, CXCL10, CXCL11, CXCL13, CXCL16; CCL2, CCL3, CCL4, CCL5, CCL8, CCL15, CCL11, CCL19, CCL20; CX3CL1	CCL1, CCL2, CCL13, CCL14, CCL17, CCL18, CCL22, CCL23, CCL24, CCL26, CCL5	search for antagonists, search for the genes encoding proteins that enhance or weaken the binding of the inducer to the receptor
Cytokines	IL-1β, IL-6, IL-12, IL-18, IL-23, TNF-α, type I IFN	IL-4, IL-13, IL-10, IL-33, TGF-β	search for the genes encoding proteins that regulate (activate or repress) the expression of effector molecules

**Table 2 cells-09-01535-t002:** The examples of M1/M2 shift in inflammatory and proliferative diseases.

Direction of the Shift	Disease
	Multiple sclerosis [38]
	Encephalomyelitis [38]
M1	Rheumatoid arthritis [39]
	Alzheimer’s disease [41] Parkinson’s disease [42]
M2	Skin leishmaniasis [43]
	Duchenne muscular dystrophy [44]
	Asthma-associated fibrosis [45]
	Hepatic cirrhosis [46]
	Pneumocystis murina infections [47]
	Various types of cancer (breast cancers, thyroid cancers, head and neck cancers, etc.) [50,51,52]

**Table 3 cells-09-01535-t003:** The main applications of macrophage reprogramming in vitro and in vivo.

Disease	Phenotype	Technology	Delivery	In Vitro/In Vivo	Results
**Inflammatory diseases Acute lung injury**	M2 to M1	Knockdown *CXCR4*	liposomes	In vitro	Inhibition of the inflammatory cytokine expression [76]
Atherosclerosis	M1 to M2	Knockdown *LAIR-1*	lentiviral vectors	In vitro	Increased foam cell formation and cholesterol uptake [29]
Vessel stroke	M1 to M2	Knockout *HDAC9*	transgenesis	In vivo	Upregulation of the lipid homeostasis genes, downregulation of the pro-inflammatory genes [31]
Myocardial infarction	M1 to M2	Overexpression of Sestrin2	lentiviral vectors	In vitro In vivo	Cardiac tissue repair [91]
Inflammation	M1 to M2	Knockout *NLRP3* (inflammasomes)	nanoparticles	In vivo	Mitigation of the acute inflammation [84]
	M1 to M2	IL-10	lentiviral vectors	In vitro	Reduction in the amounts of TNF-α, sustained macrophage polarization towards an M2 phenotype promoting local immune responses [90]
Rejection of of transplants	M1 to M2	Overexpression of Il34	AAV	In vivo	Anti-inflammatory polarization of Kupffer cells and alleviated liver transplant rejection [96]
**Proliferative diseases Ovarian cancer, lung cancer, glioma**	M2 to M1	IVT-RNA	nanoparticles	In vivo	High anti-tumor activity [23]
Melanoma, hepato- carcinoma	M2 to M1	Chloroquine	i/v	In vivo	Decreased immunosuppressive infiltration of the myeloid-derived suppressor cells and Treg cells, enhancement of the antitumor T cell-mediated immunity [24]
Ovarian cancer	M2 to M1	IFN-γ therapy	i/v	In vivo	Significant tumor regression [70]
Breast cancer, melanoma	M2 to M1	Paclitaxel	i/*p*	In vivo	Antitumor effect [25]
Breast cancer	block M2	CCL2 inhibitors	i/v	In vivo	Suppression of metastasis and prolongation of the survival [68]
Lewis lung carcinoma	block M2	MEL-dKLA	i/*p*	In vivo	Lower rates of tumor growth and angiogenesis, decreased tumor weights [27]
**Viral diseases Anti-HIV response**	block M1	iPSCT Knockout USP18	nucleofection	In vitro	Inhibition of replication of the virus [99]

i/*p*—intraperitoneal administration; i/v—intravenous administration.

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
