# Peer review of "Macrophage Modification Strategies for Efficient Cell Therapy"

_cells, 2020, doi:10.3390/cells9061535_

Round 1

Reviewer 1 Report

The manuscript touches upon an important biomedical issue and provides a good overview of potential therapeutic approaches aimed at regulating macrophage differentiation and function. Some modifications may increase the impact of the review and its potential value as a reference and educational material.

  1. More information about mechanisms by which certain factors affect macrophage differentiation is required. In particular, it remains unclear how does CXCR4 influence IL-6 and TNFα expression, how does CYBB affect M1/M2 differentiation, why HDAC9 knockout promotes the M2 phenotype.
  2. The presentation switches from discussing the M1/M2 phenotype and the ways to influence it to describing treatments unrelated to inflammation, and then getting back to M1/M2. A more consistent focus would help the reader.
  3. It remains unexplained what advantage does modification of iPSC-derived monocytes bring over direct modification of peripheral blood monocytes. 

Author Response

The manuscript touches upon an important biomedical issue and provides a good overview of potential therapeutic approaches aimed at regulating macrophage differentiation and function. Some modifications may increase the impact of the review and its potential value as a reference and educational material.

Thank you for considering our work. All your comments have been addressed, with corresponding changes made directly to the manuscript where appropriate.

More information about mechanisms by which certain factors affect macrophage differentiation is required. In particular, it remains unclear how does CXCR4 influence IL-6 and TNFα expression, how does CYBB affect M1/M2 differentiation, why HDAC9 knockout promotes the M2 phenotype.

Thank you for this point. According to your recommendation we added information for understanding the polarization pathways. Line 251;326;337.

The presentation switches from discussing the M1/M2 phenotype and the ways to influence it to describe treatments unrelated to inflammation and then getting back to M1/M2. A more consistent focus would help the reader.

Thank you for this important comment. We agree, we changed parts for understanding the line of the manuscript. 

It remains unexplained what advantage does modification of iPSC-derived monocytes brings over direct modification of peripheral blood monocytes.

Thank you for your comment. The main advantage is that a primary monocytic line is difficult to manipulate and cultivate compared with induced pluripotent stem cells. iPSC-derived monocytes could be modified and grew during the step of pluripotency. Thus, we could obtain a huge amount of cells and apply different modification methods without influence on viability.  “Induced pluripotent cells are highly proliferative, and the modified (edited) cells can be easily manipulated to produce any number of colonies of identical cells for analysis and subsequent differentiation into specific lineages.”

Reviewer 2 Report

Poltavets et al. provide a concise review of techniques available for modifying macrophages. The review is well-written and provide sufficient justification on why the topic may be of interest and relevance to readers investigating various diseases. The authors discuss the history of macrophages including more recent shifts in our understanding of macrophages. Finally, the latest advances in the technology surrounding macrophage modification strategies were addressed.

However, there are several points of improvement.

Major comments:

  • There were several points throughout the review where statements are made without any citations or references. E.g. Line 106 on monocyte heterogeneity could use a reference.
  • Key papers on macrophage ontogeny, while understandably not the focus of this review, were not cited. Line 44 does not give enough credit on extensive work done by several groups on macrophage ontogeny. See review “Ginhoux and Guilliams Immunity 2016 PMID: 26982352” for original papers on the topic.
  • I personally dislike the M1/M2 terminology used by authors as macrophages in vivo exist on a spectrum depending on the environmental stimuli but acknowledge that this dichotomy (originally based on in vitro generated cells) is still used for simplicity purposes. My suggestion is to focus on the desired functionality: i.e. pro-inflammatory or anti-inflammatory macrophages. Consider citing “Murray et al. Immunity 2014 PMID: 25035950” on nomenclature guidelines by experts in the field. However, I do not really agree with lines 72–74. If the authors claim that the ability to polarize to particular phenotypes is inherent in the cellular lineage rather than in response to the environment, they should cite any original paper to support this statement.

Minor comments:

  • Section 2.1. Are there more examples of anti-inflammatory “M2” macrophages being associated with disease severity. The one example of leishmaniasis feels unbalanced compared to the role of pro-inflammatory “M1” macrophages.
  • Section 2.2. Are checkpoint blockade therapies (PD-1 and CTLA-4) necessarily macrophage dependent? Please cite original papers if so.
  • Section 3. It is difficult to keep track if a method was used in vitro or in vivo. Authors can improve clarity while describing these studies.
  • Line 419: is it necessary to write hCD45 etc.? Probably sufficient to mention that these are human cells. On the other hand, line 39 in the intro could benefit from mentioning that these are human markers.
  • What about CAR-Macrophages recently described by “Klinchinsky et al. Nature Biotech 2020 PMID: 32361713)?

Author Response

Poltavets et al. provide a concise review of techniques available for modifying macrophages. The review is well-written and provide sufficient justification on why the topic may be of interest and relevance to readers investigating various diseases. The authors discuss the history of macrophages including more recent shifts in our understanding of macrophages. Finally, the latest advances in the technology surrounding macrophage modification strategies were addressed.

Thank you for considering our work and fair comments. We tried to address all your comments with corresponding changes made directly to the manuscript where appropriate. It is our belief that the manuscript is substantially improved after making the suggested edits.

However, there are several points of improvement.

Major comments:

There were several points throughout the review where statements are made without any citations or references. E.g. Line 106 on monocyte heterogeneity could use a reference.

Thank you. Yes, you are right; we inserted absent citations in the revised version of the Manuscript. Several statements requiring confirmation is now referenced (LINE 34, 40). We also added the historically essential works on the heterogeneity of monocytes, as you recommended (LINE 117).

Key papers on macrophage ontogeny, while understandably not the focus of this review, were not cited. Line 44 does not give enough credit on extensive work done by several groups on macrophage ontogeny. See review “Ginhoux and Guilliams Immunity 2016 PMID: 26982352” for original papers on the topic.

Thank you for this point. We expanded this part of the Manuscript by pointing out the experimental data from the paper you recommended and conclusions from reviews on the same topic. We added the following fragment:

“The modern concept of macrophages development includes three sources of these cells that correspond to three generations of hematopoietic stem cells (Elchaninov et al. 2019; Ginhoux and Guilliams 2016). The first generation develops from extra-embryonic yolk sac posterior plate mesoderm in the blood islands (Gomez Perdiguero et al. 2015; Tober et al. 2007). Evidently, these cells give rise to the microglia of the central nervous system (Hoeffel et al. 2015). The second wave of hematopoietic progenitors develops from the yolk sac hemogenic endothelium and called erythro-myeloid precursors. After establishing of blood circulation, these cells subsequently colonize the embryonic liver. Erythro-myeloid precursors could generate granulocytes, monocytes, and macrophages lineages  (Bertrand et al. 2005; Hoeffel and Ginhoux 2018). The third generation of hematopoietic cells derived from aorto-gonado-mesonephral zone endothelium and colonize the fetal liver where they establish hematopoiesis and the red bone marrow where they generate bone marrow hematopoietic stem cell lineage (Ginhoux and Guilliams 2016). During embryogenesis organs macrophages are predominantly represented by cells of the second and third generations while in the postnatal period, the percentage of macrophages derived from the third-generation rises (Perdiguero, Geissmann, and Author 2016)”.

Bertrand, Julien Y., Abdelali Jalil, Michèle Klaine, Steffen Jung, Ana Cumano, and Isabelle Godin. 2005. “Three Pathways to Mature Macrophages in the Early Mouse Yolk Sac.” Blood.

Elchaninov, Andrey V., Timur Kh. Fatkhudinov, Polina A. Vishnyakova, Anastasia V. Lokhonina, and Gennady T. Sukhikh. 2019. “Phenotypical and Functional Polymorphism of Liver Resident Macrophages.” Cells.

Ginhoux, Florent and Martin Guilliams. 2016. “Tissue-Resident Macrophage Ontogeny and Homeostasis.” Immunity 44(3):439–49.

Gomez Perdiguero, Elisa, Kay Klapproth, Christian Schulz, Katrin Busch, Emanuele Azzoni, Lucile Crozet, Hannah Garner, Celine Trouillet, Marella F. De Bruijn, Frederic Geissmann, and Hans Reimer Rodewald. 2015. “Tissue-Resident Macrophages Originate from Yolk-Sac-Derived Erythro-Myeloid Progenitors.” Nature.

Hoeffel, Guillaume, Jinmiao Chen, Yonit Lavin, Donovan Low, Francisca F. Almeida, Peter See, Anna E. Beaudin, Josephine Lum, Ivy Low, E. Camilla Forsberg, Michael Poidinger, Francesca Zolezzi, Anis Larbi, Lai Guan Ng, Jerry K. Y. Chan, Melanie Greter, Burkhard Becher, Igor M. Samokhvalov, Miriam Merad, and Florent Ginhoux. 2015. “C-Myb(+) Erythro-Myeloid Progenitor-Derived Fetal Monocytes Give Rise to Adult Tissue-Resident Macrophages.” Immunity 42(4):665–78.

Hoeffel, Guillaume and Florent Ginhoux. 2018. “Fetal Monocytes and the Origins of Tissue-Resident Macrophages.” Cellular Immunology.

Perdiguero, Elisa Gomez, Frederic Geissmann, and Nat Immunol Author. 2016. “Development and Maintainance of Resident Macrophages HHS Public Access Author Manuscript.” Nat Immunol.

Tober, Joanna, Anne Koniski, Kathleen E. McGrath, Radhika Vemishetti, Rachael Emerson, Karen K. L. De Mesy-Bentley, Richard Waugh, and James Palis. 2007. “The Megakaryocyte Lineage Originates from Hemangioblast Precursors and Is an Integral Component Both of Primitive and of Definitive Hematopoiesis.” Blood.

I personally dislike the M1/M2 terminology used by authors as macrophages in vivo exist on a spectrum depending on the environmental stimuli but acknowledge that this dichotomy (originally based on in vitro generated cells) is still used for simplicity purposes. My suggestion is to focus on the desired functionality: i.e. pro-inflammatory or anti-inflammatory macrophages. Consider citing “Murray et al. Immunity 2014 PMID: 25035950” on nomenclature guidelines by experts in the field. However, I do not really agree with lines 72–74. If the authors claim that the ability to polarize to particular phenotypes is inherent in the cellular lineage rather than in response to the environment, they should cite any original paper to support this statement.

Thank you for this important comment, we agree. The M1/M2 paradigm was established based on extreme in vitro polarization of macrophages with IFN/LPS (for M1 phenotypes) or type2-cytokines for M2 macrophage polarization. Although this concept has been very helpful, it is now established that such conceptualization cannot be applied in vivo as it does not exist due to the remarkable plasticity and heterogeneity of macrophages. Indeed, M1/M2 paradigm was based on in vitro studies, but this terminology has entered the scientific language and is often used in research. Also, attempts to combine in vitro and in vivo studies are constantly making. Based on this, more modern variants of the M1/M2 concept were proposed, and the specific signs of the M1 or M2 macrophage phenotype were clarified (Murray et al., 2014; Malyshev and Malyshev, 2015). However, an example of the usage of the M1/M2 concept in vivo studies could be found in works where the role of macrophages of different origins in liver repair was elucidated. In these studies, they suggested that migrating macrophages of monocytic origin under the action of the organ microenvironment polarize into M2 anti-inflammatory macrophages (Holt, Cheng, and Ju, 2008; Zigmond et al., 2014).

For more accuracy, we replaced the M1/M2 abbreviation in a number of in vivo studies to pro-inflammatory/anti-inflammatory and added the citation to a work of Murray et al.,(LINE 67). We also added the fragment: “However, the M1/M2 paradigm conveniently reflects the most phenotypically distant (polar) differentiation states of macrophages and this terminology has entered the scientific language and is often used in research. For in vivo studies M1/M2 dichotomy is suggested to be replaced by the terms pro-inflammatory/pro-regenerative, respectively”. (LINE 84)

Concerning the inheritance of the phenotype, we referred to the recent work of Specht and colleagues. This mostly technical-oriented paper revealed interesting features of macrophages of the same cell culture. Using single-cell proteomics, they found that in the absence of polarizing stimuli, the macrophage population existed in a continuum, showing loss or gain of proteins identified as enriched in M1 or M2 macrophages. To make the conclusion more accurate, we deleted the sentence about the inheritance of the phenotype because the authors actually didn`t find the experimental confirmation and just presented it as an assumption.  

Holt, M. P., Cheng, L. and Ju, C. (2008) ‘Identification and characterization of infiltrating macrophages in acetaminophen-induced liver injury’, Journal of Leukocyte Biology. Wiley, 84(6), pp. 1410–1421. doi: 10.1189/jlb.0308173.

Malyshev, I. and Malyshev, Y. (2015) ‘Current Concept and Update of the Macrophage Plasticity Concept: Intracellular Mechanisms of Reprogramming and M3 Macrophage "Switch" Phenotype.’, BioMed research international, 2015, p. 341308. doi: 10.1155/2015/341308.

Murray, P. J. et al. (2014) ‘Macrophage Activation and Polarization: Nomenclature and Experimental Guidelines’, Immunity, 41(1), pp. 14–20. doi: 10.1016/j.immuni.2014.06.008.

Zigmond, E. et al. (2014) ‘Infiltrating Monocyte-Derived Macrophages and Resident Kupffer Cells Display Different Ontogeny and Functions in Acute Liver Injury’, The Journal of Immunology, 193(1), pp. 344–353. doi: 10.4049/jimmunol.1400574.

  Minor comments:

Section 2.1. Are there more examples of anti-inflammatory “M2” macrophages being associated with disease severity. The one example of leishmaniasis feels unbalanced compared to the role of pro-inflammatory “M1” macrophages.

Thank you for this note. We added the information about M2 shift in Duchenne muscular dystrophy, asthma-associated fibrosis, liver cirrhosis and Pneumocystis murine infection (LINES: 164-170).

Section 2.2. Are checkpoint blockade therapies (PD-1 and CTLA-4) necessarily macrophage dependent? Please cite original papers if so.

Thank you for your question. The anti-PD-1 and anti-CTLA-4 checkpoint blockade therapy firstly was described as T-cell dependent, but last studies showed that these therapies may also function through a direct effect on macrophages, with substantial implications for the treatment of cancer. We cited original papers in the manuscript (LINE 195).

Section 3. It is difficult to keep track if a method was used in vitro or in vivo. Authors can improve clarity while describing these studies.

Yes, you are right, some studies need to be described preciser, we revised them (LINES 324, 334, 345, 354, 385, 399).

Line 419: is it necessary to write hCD45 etc.? Probably sufficient to mention that these are human cells. On the other hand, line 39 in the intro could benefit from mentioning that these are human markers.

Thank you for this note. We removed “h” prefix and mentioned the organism (LINE 503). We also pointed out that mentioned markers are human (LINE 41) as you recommended.

What about CAR-Macrophages recently described by “Klinchinsky et al. Nature Biotech 2020 PMID: 32361713)?

Thank you for this addition. It is useful and actual information which now is added in the manuscript (LINE 558).

Reviewer 3 Report

The authors provide a comprehensive overview of the macrophage modification strategies. After a general introduction, the authors introduced macrophage-related inflammatory and proliferative diseases. Later they focused on reprogramming strategies and finally on M1/M2 cells in experimental clinical research.

In my opinion, the article is not very well structured, and it lacks informative/detailed tables.  This article is rather interesting for uninformed readers and does not include much new information. It may benefit from additional tables, for example, macrophage dependent diseases table, macrophage-reprogramming strategy table. In my opinion, the information in the tables should be detailed and include novel findings and references. Tables are an effective way to present large amounts of data. Unfortunately, in the manuscript, the only table is basic and includes only well-known information; “search tactic” does not include practically any information and reference.  Besides, a review-figure could be used to communicate complex information that would be complicated to explain in text. The figure in the manuscript is a figure for non-expert audiences.

Author Response

The authors provide a comprehensive overview of the macrophage modification strategies. After a general introduction, the authors introduced macrophage-related inflammatory and proliferative diseases. Later they focused on reprogramming strategies and finally on M1/M2 cells in experimental clinical research.

In my opinion, the article is not very well structured, and it lacks informative/detailed tables.  This article is rather interesting for uninformed readers and does not include much new information. It may benefit from additional tables, for example, macrophage dependent diseases table, macrophage-reprogramming strategy table. In my opinion, the information in the tables should be detailed and include novel findings and references. Tables are an effective way to present large amounts of data. Unfortunately, in the manuscript, the only table is basic and includes only well-known information; “search tactic” does not include practically any information and reference.  Besides, a review-figure could be used to communicate complex information that would be complicated to explain in text. The figure in the manuscript is a figure for non-expert audiences.

Thank you for interest in our work and for constructive and helpful comment that will greatly improve the manuscript. We prepared the table of reprogramming approaches, macrophage dependent diseases and strategies for their resolution. Besides, we agree with you about Figure and suggest it as a graphical abstract.

Round 2

Reviewer 3 Report

Unfortunately, the authors did not addressed most of my comments.

In my opinion, the article is still not very well structured, and it lacks informative/detailed tables. The table1 still does not include novel findings and references. The new table2 improved the paper but it is still not an example of a table that organize the literature in a review article.

Overall, the authors just made some minor changes.

Author Response

Dear Reviewer,

Thank you for your attention to our work and apologize for the insufficient changes made in the first round of the review.

We radically changed the structure of the manuscript, according to your recommendations. Now it begins with the «Macrophage development» section, then comes the part on the M1/2 paradigm and it is accompanied by a Table 1, which was improved according to more recent articles. Table 1 is also used to clarify the section concerning the reprogramming strategy. "The role of the monocyte-macrophage system in disease" is accompanied by a new short table reflecting the M1/2 shift in various disorders. Table 3 is devoted to the main applications of macrophage reprogramming and it reflects both the data listed in the sections on M1/M2 clinical and experimental applications and the results of various macrophage reprogramming strategies. We should note that the section «The existing approaches for macrophage reprogramming» is actually complicated, however, in its current form it is structured as much as possible and is divided into information about the acting agents, methods of a transient or permanent modification, methods of delivering the vector, iPSC as a separate area and in vivo models. More simplification is impossible because often, the methods of reprogramming macrophages and how the construct is delivered are the same in most cases. The method of exposure (permanent or transient) is mostly determined by the method of delivery of the vector/signal molecule (AAV, lentiviruses, lipocomplexes).

Thus, each significant part of our review is accompanied by a table designed to systematize the information presented. We thank you for your recommendations and hope you will find the changes sufficient.

Round 3

Reviewer 3 Report

I do not have any further comments

Author Response

Dear Reviewer,

Thank you for considering our work and fair comments.